# Development and Validation of a Questionnaire of the Perioperative Nursing Competencies in Patient Safety

**DOI:** 10.3390/ijerph19052584

**Published:** 2022-02-23

**Authors:** Ester Peñataro-Pintado, Encarna Rodríguez-Higueras, Mireia Llauradó-Serra, Noelia Gómez-Delgado, Rafael Llorens-Ortega, José Luis Díaz-Agea

**Affiliations:** 1Nursing Department, University School of Nursing and Occupational Therapy of Terrassa (EUIT), 08221 Terrassa, Spain or epenataro@uic.es (E.P.-P.); rafallorens@euit.fdsll.cat (R.L.-O.); 2Nursing Department, Campus Sant Cugat, International University of Catalonia (UIC), 08195 Sant Cugat del Vallès, Spain; mllaurados@uic.es; 3Nursing Department, Vall d’Hebron University Hospital, 08035 Barcelona, Spain; nogomez@vhebron.net; 4Nursing Department, Catholic University of Murcia (UCAM), 30107 Guadalupe de Maciascoque, Spain; jluis@ucam.edu

**Keywords:** perioperative nursing, competencies, patient safety, Delphi method, questionnaire, psychometric properties

## Abstract

(1) Background: This research presents the CUCEQS© (Spanish acronym for Questionnaire of Perioperative Nursing Safety Competencies), which evaluates the perception of perioperative nurses about their competencies related to surgical patient safety. The aim of the present study was to design, validate, and analyze the psychometric properties of the CUCEQS©. (2) Methods: We devised an instrumental, quantitative, and descriptive study divided into two phases: in the first, the questionnaire was designed through a Delphi method developed by perioperative nurses and experts in patient safety. In the second, the reliability, validity, and internal structure of the tool were evaluated. (3) Results: In the first phase, the items kept were those that obtained a mean equal to or higher than four out of five in the expert consensus, and a Content Validity Index higher than 0.78. In the second phase, at the global level, a Stratified Cronbach’s Alpha of 0.992 was obtained, and for each competency, Cronbach’s Alpha values between 0.81 and 0.97 were found. A first-order confirmatory factor analysis of the 17 subscales (RMSEA 0.028, (IC 90% = 0.026–0.029) and its observed measures was performed for the 164 items, as well as a second-order analysis of the four competencies (RMSEA = 0.034, (IC90% = 0.033–0.035). (4) Conclusions: The questionnaire is a valid tool for measuring the perceived level of competency by the perioperative nurses in surgical patient safety. This is the first questionnaire developed for this purpose, and the results obtained will facilitate the identification of areas to be improved by health professionals in patient safety

## 1. Introduction

Assuming that to err is human [1] is the first step to try to avoid making mistakes. The second step is to analyze the personal factors and the system that contribute to this. Evidence [2,3,4,5] reveals that about 25% of patients have some kind of postoperative complication, with 3 to 16% of these complications being major complications. Between 3 and 22% of the adverse events lead to disability, with mortality rates ranging from 0.4 to 0.8%, and with around 50% of these events being avoidable.

The most frequent errors are related to surgical procedures, the location of surgery, laterality, or patient errors [6,7,8]. The etiology of the errors is diverse, but the most important ones are: not following the safety guidelines, communication problems within the multidisciplinary team, and a lack of leadership [9].

At present, the use of a surgical verification list has significantly reduced the complications and mortality rates [10]. The checklist, a verification tool created by the WHO, focuses on five specific areas of the perioperative process: correct marking of the surgical area; the establishment of systems to avoid medication errors related to anesthesia; correct and systematic counting of gauzes; the development of effective communication; and the minimization of interruptions in a cohesive surgical team. An observational study [11] showed a reduction in surgical complications from 11 to 7%, mortality from 1.5 to 0.8%, and an increase in the compliance of safety measures from 34 to 56% [11]. However, the literature has also indicated that for the verification list to be useful, the organizations and their teams must make significant changes. These changes must be associated with the institutional culture to provide coherence on the use of the verification list. This is a way in which to rethink the flow of work, communication and leadership in the processes [12,13,14,15,16].

The perioperative nurse is found in a multidisciplinary team and is the only health professional who accompanies the patient throughout the entire perioperative process. The present study focuses on the nursing discipline and perioperative nursing competencies, originally defined by the EORNA (European Operating Room Nurses Association) in 1997, and which were included and developed in the EORNA Common Core Curriculum for Perioperative Nursing published in 2012 and 2019 [17].

Therefore, with this guide in mind, we developed the competencies, sub-competencies, and the items from each of the competency areas in a detailed manner, steering them towards clinical safety, considering scientific evidence and the criteria of experts.

## 2. Materials and Methods

### 2.1. Hypothesis

The CUCEQS© (in Spanish: “*CUestionario de Competencias de la Enfermería Quirúrgica en Seguridad*”) is a questionnaire which has correct psychometric properties of validity and reliability. It is to be used in the Spanish context as an instrument for measuring the competency levels perceived in perioperative nursing, for the clinical safety of the surgical patient.

### 2.2. Aim

The aim of the study is the development and validation of the CUCEQS©, which measures the level of competency perceived by perioperative nursing related to surgical patient safety.

### 2.3. Design

This study is an instrumental, quantitative, and descriptive study for the design and validation of a questionnaire. This research followed the STROBE checklist (Appendix A ).

The study was divided into two phases: Phase I—development of the tool, and Phase II—validation of the tool.

### 2.4. Participants

#### 2.4.1. Phase I—Development of the Tool

This phase was carried out using a modified Delphi Method, with the implementation of three rounds.

In the first round, the theoretical framework of the EORNA was adopted [17], and five focus groups were developed with perioperative nurses from Spain, which dealt with key aspects of surgical patient safety [18] between June 2017 and February 2018.

Starting with the analysis of the focus groups, the research team designed the CUCEQS© questionnaire. For definition of the tool, its components (competencies, sub-competencies, and items), and its structure, the competency guide proposed by European and American perioperative societies [17,19,20], as well as the national and international patient safety strategic line^4^, were used as references.

Afterwards, two rounds were performed with an expert panel to obtain consensus, between April and June 2018.

The experts had to comply with the following criteria:

Perioperative nurses: (a) with more than 5 years of continuous work experience, mostly full-time, in operating rooms or postoperative units at second- or third-level public hospitals in Spain, (b) who were currently employed, and (c) who had a minimum specific postgraduate education in perioperative nursing. They were contacted through their supervisors.

Experts in clinical safety: (a) who were part of the management board of institutions or associations, or were members of safety commissions at their workplace, with a broad public and/or scientific trajectory in patient safety. They were contacted via email, which was provided to us or publicly available.

The selection of the expert panel was intentional, and 40 individuals from Spain were initially contacted [21]; 38 experts agreed to participate, and 37 finished both rounds. Of these, 13 were clinical safety experts and 25 were perioperative nurses. As for the demographic characteristics of the experts, 89.5% were women and 10.5% were men. Their average age was 46.7 years (SD = 7.5) (there is detailed information about the experts in Appendix A).

The rounds were structured, with a controlled and anonymous online interaction. For each round, a length of time of 4 weeks was set, and many reminders were sent to obtain the maximum number of responses. Finally, three months were needed to finish the entire process. The experts had to evaluate the importance of each item using a Likert scale, which ranged from 1, “not very adequate”, to 5, “very adequate”; they could include comments about the readability or the drafting of the items or add new ones.

#### 2.4.2. Phase II—Validation of the Tool

Throughout 2019, in this phase, the sample was obtained due, in part, to the personalized contact of the main researcher with the coordinators of the operating room from the hospitals in the geographic area and surroundings. The questionnaires were provided as hard copies and collected after 4 weeks. Also, the description of the study was disseminated through social networks associated with perioperative nursing and through email, to obtain the greatest dissemination in Spain. To allow for anonymous surveys, non-traceable links were used.

To analyze the validity of the criteria, nurses were selected who were experts in clinical safety, and who were students enrolled in the Masters in Perioperative Nursing program, who had received specific training on the safety of surgical patients from Spain.

The target sample size was estimated to test the hypothesis, with 164 variables in a first-order confirmatory model with 17 factors and 4 second-order factors; the Root-Mean-Square error of approximation (RMSEA) was used, with a population value of 0.08 and an alternative hypothesis RMSEA of 0.05, using nominal alpha 0.05 and 90% power. Based on these assumptions, the target sample size needed was 398 subjects, according to the model to be tested and the number of factors to be determined.

### 2.5. Data Analysis

In the first phase, the analysis of data was performed with the SPSS statistical program version 22 (Statistical Package for the Social Sciences, IBM Corp. Armonk, NY, USA). The data were obtained through analysis of the means and standard deviation of each item and the sub-competency of the questionnaire, with the following results considered adequate: a mean > 4 out of a maximum value of 5, and a Content Validity Index (CVI) > 0.78 [22]. Also, qualitative assessments of the readability of the items and the structure of the questionnaire itself were solicited. The validity of the questionnaire was obtained with these results.

In the second phase, the psychometric characteristics of the questionnaire were analyzed. The overall internal consistency of the questionnaire was evaluated with Stratified Cronbach’s Alpha. For each of the competencies and sub-competencies, Cronbach’s Alpha was calculated for both the test and the retest. Thus, the homogeneity of the statements was measured, indicating a relationship between them. The temporal stability of the testretest was evaluated with the Intraclass Correlation Coefficient (ICC).

To validate the criteria, we analyzed the degree of agreement between the perioperative nurses and nine expert nurses in clinical safety. The median and the interquartile range were calculated for each competency and sub-competency, and comparisons were made between the two groups with a Mann–Whitney non-parametric U test.

In this same section, to increase the validity of the criteria, we analyzed the level of agreement among the responses from a group of Perioperative Nursing Master’s students. The results of two groups were compared, one of which had received specific training on surgical patient safety, and another that had not received such training.

Given that the structure of the questionnaire was theoretically derived and designed specifically for the measurement of self-perception, the most appropriate analysis that could be used to obtain the validity of the construct was a confirmatory factor analysis. For this, the lavaan package of R, a language and environment for statistical computing, was utilized (version 3.6.1) [23]. The confirmatory analysis represents the measurement model which describes the associations between the latent variables.

Two confirmatory factor analyses were performed: the analysis of the 17 first-order subscales and their measurements observed in the 164 items, and an analysis of the four second-order competencies. Given that the items were ordinal categories, the model was adjusted with a Categorical Item Factor Analysis; this was used on the matrix of tetrachoric correlations, through the use of an estimator of unweighted least squares (ULS), and adjusted according to means and variances for a robust estimation of non-normality. The model fit was calculated with the Root-Mean-Square Error of Approximation (RMSEA) method, considering the values less than 0.05 as acceptable [24], and Standardized Root-Mean-Square Residual (SRMR). The factorial loads and the relative adjustment were analyzed through the Comparative Fit Index (CFI) and the Tucker–Lewis Index (TLI), considering values ≥0.90 as acceptable [25].

It is worth highlighting that an exploratory factorial analysis was performed, but it classified the structure of factors and variables differently and mainly without sense. The questionnaire followed the order of the perioperative process, according to the European Operting Room Nurse Association (EORNA, 2009; EORNA, 2019), and also included the main activities associated with the safety of the patient according to nursing roles. The classification of the variables into another category did not make sense. Thus, the exploratory factorial analysis was discarded. For example, in some evaluated factors, an item related to the sterile surgical field was assigned to a member of the surgical team who was not part of the sterile team.

### 2.6. Ethical Aspects

The study was approved by the Research Ethics Committees from the different centers that participated in the study. The participants were informed that the data were confidential and anonymous (General Regulation of Data Protection (RGPD) from 2018).

All the participants in the study voluntarily accepted the invitation to participate, and anonymity and confidentiality of the data were guaranteed. Only the responses from the nurses who participated in the test retest were codified to be able to analyze the data, and posteriorly anonymized.

The authorization of the professionals who comprised the panel of experts was solicited for the inclusion of their data in the present research study.

## 3. Results

The main result of the present study was the creation of the CUCEQS© questionnaire. It measures the level of competency in surgical patient safety perceived by the perioperative nurses (Appendix A CUCEQS© questionnaire).

### 3.1. Phase I—Development of the Tool

The stages in which the Delphi method was developed were the following (Figure 1):

#### 3.1.1. Preparatory Phase

Starting with the work by the EORNA, and after the focus groups of round 1 [19] the competency areas and the first version of the questionnaire were obtained; it was comprised of 4 competencies, 17 sub-competencies, and 163 items.

#### 3.1.2. Consultation Phase

Two more rounds were implemented. In the first one, the participation of the experts was 100%, and in the second one, this decreased to 97.4%. A consensus between experts was reached for the sub-competencies and the items. In both rounds, the consensus mean was equal to or higher than 4, with a CVI of 0.78 (the scores can be found in Appendix A).

In both rounds, proposals were provided about the distribution and improvements to the writing of the sub-competencies and the items. In fact, an item was split into two, which resulted in a final number of 164 items.

#### 3.1.3. Phase of Consensus

The final version of the questionnaire was divided into 4 competencies in agreement with the EORNA, and into 17 sub-competencies and 164 items.

The competency areas and the competencies of the final version of the CUCEQS© are shown in Table 1. For more details, please go to the Appendix A (CUCEQS©).

The global questionnaire had a different range of scores depending on the role of the perioperative nurse (scrub nurse, circulating nurse, anesthesia nurse or postoperative nurse). Also, the score of each competency and sub-competency could vary (Appendix A). Given the importance of this, the questionnaire provides a brief explanation at the beginning for completing the questionnaire correctly.

### 3.2. Phase II—Validation of the Tool

Phase II consisted of the participation of 415 Perioperative nurses from 55 hospitals, 9 nurses who were experts in clinical safety, and 56 students enrolled in their postgraduate studies in perioperative nursing at two universities. The sociodemographic characteristics of the sample were analyzed (Table 2).

From the total sample of perioperative nurses, 109 performed the test retest after 3 weeks. These data were used to calculate and compare the medians of the test and retest for each competency and for each sub-competency (Appendix A), as well as the psychometric properties of the questionnaire.

#### 3.2.1. Reliability

As for the internal consistency, the overall reliability of the questionnaire was excellent, as shown by the Stratified Cronbach’s Alpha of 0.992 found for the test, and the Stratified Cronbach’s Alpha of 0.985 found for the test retest.

For the test, each competency obtained a Cronbach’s Alpha between 0.81 and 0.97. Each sub-competency obtained a Cronbach’s Alpha between 0.75 and 0.99.

In the test-retest phase, a Cronbach’s Alpha was obtained between 0.84 and 0.96 at the level of competency, and between 0.66 and 0.98 at the sub-competency level (Table 3).

For the analysis of temporal stability of the questionnaire, the Intraclass Correlation Coefficient (ICC) was calculated. All the values obtained were equal to or higher than 0.77, which indicated good reliability (Appendix A).

#### 3.2.2. Validity

The validity of the criteria was also calculated. For this, the scores from the Perioperative nurses and the nurses who were experts on clinical safety were compared to analyze the level of agreement between the two groups. Significant results were obtained, which demonstrated that 100% of the experts in clinical security obtained scores that were equal to or higher than the perioperative nurses (Appendix A)

The nurses who were specialists in patient safety manifested that their lack of experience in the area of surgery impeded them from answering some items from the questionnaire.

To re-enforce the validity of the criteria, an analysis was performed of the scores of the Perioperative Nursing Master’s students who had received specific training on the safety of surgery patients, with respect to those from the same year who had not received this training. Significant results were found, and it was observed that those who had received the training obtained scores that were equal to or higher than the non-trained group (Appendix A).

On the other hand, to obtain the validity of the construct, a confirmatory factor analysis was performed, where it was observed that the CUCEQS© model had two different levels. In the first order, we found the 17 subscales from C1C1 to C4C4.

In the second order, we found the four subscales, from C1a to C4 (Figure 2).

The first-order model showed an excellent fit with χ2/degrees of freedom of 1.39 (χ^2^ = 17.194.64, df = 13.066). Likewise, it obtained an excellent approximate fit value (RMSEA = 0.028, (CI90% = 0.026–0.029); SRMR 0.081), and relative fit value (CFI = 0.985, TLI = 0.985). The model of second-order competency areas also showed excellent fit values, with a χ^2^/degrees of freedom ratio of 1.49 (χ^2^ = 19.65680; df = 13.179), and the approximated fit value of RMSEA = 0.034, (CI90% = 0.033–0.035; CFI = 0.977; TLI = 0.976). The model-fit results support the theoretical structure established for the questionnaire and its measurement model.

## 4. Discussion

The CUCEQS© questionnaire was shown to have very robust psychometric properties for measuring the perception of perioperative nurses on the safety of surgery patients in their usual practice at work. Also, a detailed analysis of the instrument was reported to promote its use. Thus, we are making a contribution to the previous work by the EORNA [18,21] through the creation of a questionnaire that was designed by perioperative nurses and experts in the area of safety.

The results from phase I of the present study showed that reliable evaluation and consensus between the experts determined the suitability of the theoretical competencies, the sub-competencies, and the targeted items on the safety of the surgical patient. The complexity of the subject resulted in the panel of experts being comprised of perioperative nurses, as well as experts on the safety of patients. This nexus was necessary for dealing with all the dimensions of perioperative safety.

During the design phase, the questionnaire obtained absolute consensus from the experts. The structure and the validity of the construction of the questionnaire showed two models with an excellent fit.

The CUCEQS© tool has 17 sub-competencies which make up the 4 competencies and the competency areas of which it is comprised.

The ethical and legal competency is formed of two sub-competencies that include the most basic aspects on the application of safety standards, including the safe-surgical checklist. The items are congruent with the literature, as they come from different safety standards [2,5,14]. Competency 2 is associated with perioperative care. It is comprised of eight sub-competencies. These describe the different roles that the perioperative nurse can play in each phase of the surgical process (scrub nurse, circulating nurse, anesthesia nurse and postoperative nurse). All the Cronbach values were excellent, except for the sub-competency associated with thermoregulation. There is a strong agreement between the guides, the protocols and the health professionals on the aspects related to the care of surgical patients in each process, except for the treatment of thermoregulation; thus, patients continue experiencing perioperative hypothermia [26]. The questionnaire includes the most evident items, but there is a great variability in protocols for treating thermoregulation [27,28,29,30]. This is be a clear aspect that could be worked on and agreed upon as a team. There is evidence which shows that surgical hypothermia in the patient is a precursor to postoperative complications, such as alterations in coagulation or pain [30,31]. It is important to establish strategies for measurement and prevention, and the CUCEQS© questionnaire will help perioperative nurses to become aware of the link between surgical thermoregulation and the safety of the patient [26,28,32].

Competency 3, related with efficient communication, is composed of three sub-competencies that include interpersonal relations with the patient, the family, and the surgical team itself. The evidence shows the direct relationship that exists between communication and the safety of the patient [33,34]. The main causes of the mistakes that are produced are the interruptions in communication [35]. This finding reflects the importance of working and creating strategies for improving this aspect, and the development of good nursing leadership within a multidisciplinary team is highlighted.

Lastly, competency 4 is related with the culture of safety, and is composed of four sub-competencies. This competency was the most underlined by the experts who participated in the Delphi method. In their qualitative contributions, the experts highlighted the importance of professionals’ training on the culture of safety; however, they also mentioned that in the reality of caregiving, this is not provided, and care-related activities are prioritized with respect to the quality of care or the safety of the patient. Therefore, strategies are needed to train and develop this culture of safety in organizations, especially in aspects related to communication and teamwork [36]. The sub-competency related with the notification of errors was highlighted, as it obtained a Cronbach’s Alpha of 0.78. This is a good value, but with respect to the competency to which it belonged, it is lower, as this competency obtained a Cronbach’s Alpha value of 0.92, which is an excellent value. This difference in scores could be related to the difficulty in recognizing when mistakes have been made and considering them as a learning opportunity to improve the safety of the patient [2,36,37,38]. This competency also includes the sub-competency associated with the interprofessional knowledge and limitations, as nurses must be highly up-to-date, and must also resort to scientific evidence to maintain their criteria and to anticipate the needs that could arise [17]. If we consider CUCEQS© in its totality, it has an excellent global reliability with a Stratified Cronbach’s Alpha of 0.992.

Thus, we can consider that the questionnaire will help nurses to reflect on and be- come aware of their weaknesses and strengths. Moreover, the professional could take both individual and team actions, including their managers, that could help to increase the safety of the patient, which could result in the reduction in adverse events and errors. The results obtained on the internal consistency of the questionnaire are very favorable if we compare them with other questionnaires [39,40]. These data compel us to create a reduced version that is specially oriented towards the health professional who is already entrenched in the perioperative area. For novel professionals, having such a detailed questionnaire available could help with their training, and could even become a standardized and reliable evaluation tool to be considered by trainers.

The questionnaire was shown to be sensitive to changes, and different scores were obtained on training related to safety, depending on the four profiles of the professionals who collaborated on the attainment of the validity of the criteria. The questionnaire’s sensitivity to change is fundamental for its future use and for evaluating the effectiveness of professionals’ training in this area of knowledge. Reliable measurement indicators are needed, and the CUCEQS© questionnaire is reliable for the area of perioperative nursing [41].

## 5. Strengths, Limitations, and Areas for Further Research

The greatest strength of the research is to have a validated tool that collects the competencies of perioperative nursing. There is no previous tool that has the same characteristics.

As regards limitations, we can consider that the CUCEQS© questionnaire is perhaps too long, as it considers the entire Perioperative process and all the perioperative nursing roles.

Another limitation was the small sample found for the groups of nurses who were experts in clinical safety, and who were utilized to contrast the validity of the questionnaire criteria.

## 6. Conclusions

The results show that we have designed a tool that measures the level of competency perceived by perioperative nurses in relation to the safety of the surgical patient. This tool is meaningful for this purpose, and its validity and reliability values are very consistent. We underline the absence of scales or instruments of these characteristics in the area of surgery or the operating room.

## Figures and Tables

**Figure 1 ijerph-19-02584-f001:**
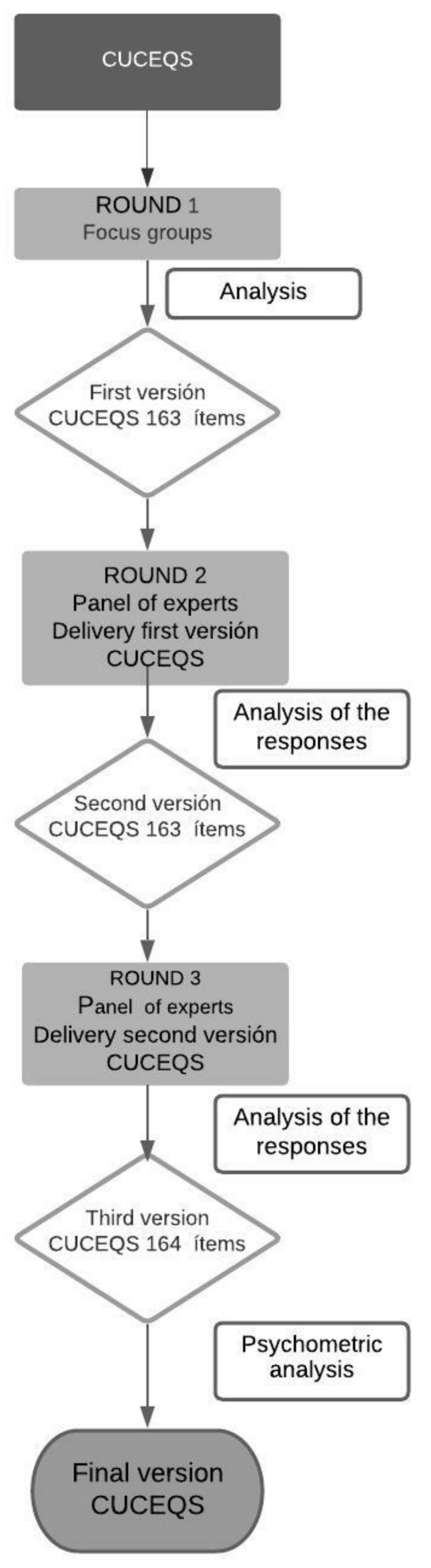
This is a figure of the procedure for the design of the CUCEQS© questionnaire.

**Figure 2 ijerph-19-02584-f002:**
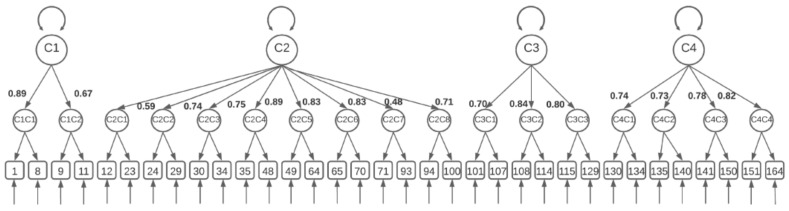
This is a figure of the confirmatory factor analysis: C1: ethical and legal practice; C2: perioperative care; C3: communication; C4: culture of safety. C1C1: safety standards; C1C2: surgical checklist; C2C1: anesthesiology nurse; C2C2: safe placement of the patient; C2C3: surgical thermoregulation; C2C4: circulating nurse; C2C5: scrub nurse; C2C6: safe use of the electric scalpel; C2C7: postoperative nurse; C2C8: postoperative pain; C3C1: efficient communication with the user; C3C2: communication and teamwork; C3C3: leadership; C4C1: professional culture of safety; C4C2: institutional culture of safety; C4C3: errors; C4C4: knowledge.

**Table 1 ijerph-19-02584-t001:** This is a table of the competency areas and competencies of the final version of the CUCEQS© questionnaire.

Competency Area	Competency
Ethical and legal practice	C1. Exerted in agreement with the legislation, ethics, and professional orientation within the area of perioperative nursing.
Perioperative care	C2. Provides perioperative nursing care integrating evidence-based knowledge and practice into a safe environment.
Interpersonal relations/ communication	C3. Establishes and maintains effective interpersonal relationships with the patients and surgical team during the Perioperative process.
Culture of safety	C4. Promotes the safety culture of the surgical patient.

**Table 2 ijerph-19-02584-t002:** This is a table of the sociodemographic characteristics of the perioperative nurses, Perioperative Nursing Master’s students, and nurses who are experts in safety.

		Perioperative Nurses (N = 415)		Perioperative Nursing Master’s Students (N = 56)		Safety Expert Nurses (N = 9)	
		N	% Median [Interquartile Range]	N	% Median [Interquartile Range]	N	% Median [Interquartile Range]
Sex	Male	53	12.8%	12	21.4%	1	11.1%
	Female	362	87.2%	44	78.6%	8	88.9%
Age			40.0 [32.0;47.0]	56	28 [25.7;32.5]		47.0 [42.0;51.0]

Number of years in the nursing profession			17.0 [8.00;23.0]	56	3.5 [1;6.2]		14.13 [10.00;17.00]
Number of years as a perioperative nurse			12.0 [4.00;19.0]	56	0 [0;1]		0
Specific training of the safety of the surgical patient		0	0	56	53.75%	9	100%

**Table 3 ijerph-19-02584-t003:** This is a table of the Cronbach’s Alpha according to competency and sub-competency.

	Cronbach’s Alpha Test	Cronbach’s Alpha Test Retest
C1EC1	0.746 [0.631;0.814]	0.789 [0.628;0.863]
C1EC2	0.796 [0.741;0.843]	0.804 [0.697;0.875]
C1 TOTAL	0.802 [0.714;0.855]	0.84 [0.735;0.896]
C2EC1	0.967 [0.952;0.977]	0.91 [0.742;0.955]
C2EC2	0.886 [0.823;0.92]	0.806 [0.725;0.857]
C2EC3	0.765 [0.671;0.825]	0.662 [0.51;0.757]
C2EC4	0.966 [0.946;0.976]	0.908 [0.756;0.958]
C2EC5	0.986 [0.977;0.991]	0.805 [0.699;0.879]
C2EC6	0.936 [0.901;0.956]	0.683 [0.551;0.773]
C2EC7	0.996 [0.995;0.997]	0.983 [0.945;0.992]
C2EC8	0.962 [0.949;0.972]	0.884 [0.831;0.912]
C2 TOTAL	0.977 [0.974;0.98]	0.963 [0.942;0.973]
C3EC1	0.931 [0.893;0.952]	0.947 [0.868;0.973]
C3EC2	0.885 [0.838;0.918]	0.889 [0.815;0.93]
C3EC3	0.916 [0.899;0.932]	0.938 [0.905;0.963]
C3 TOTAL	0.941 [0.928;0.951]	0.956 [0.939;0.968]
C4EC1	0.893 [0.87;0.911]	0.889 [0.851;0.917]
C4EC2	0.91 [0.894;0.927]	0.922 [0.89;0.944]
C4EC3	0.789 [0.76;0.813]	0.806 [0.756;0.848]
C4EC4	0.845 [0.82;0.868]	0.855 [0.812;0.886]
C4 TOTAL	0.927 [0.915;0.936]	0.945 [0.928;0.955]

C1: ethical and legal practice; C2: perioperative care; C3: communication; C4: culture of safety. C1EC1: safety standards; C1EC2: surgical checklist; C2EC1: anesthesiology nurse; C2EC2: safe placement of the patient; C2EC3: surgical thermoregulation; C2EC4: circulating nurse; C2EC5: scrub nurse; C2EC6: safe use of the electric scalpel; C2EC7: postoperative nurse; C2EC8: postoperative pain; C3EC1: efficient communication with the user; C3EC2: communication and teamwork; C3EC3: leadership; C4EC1: professional culture of safety; C4EC2: institutional culture of safety; C4EC3: errors; C4EC4: knowledge.

## Data Availability

Data are available by contacting the corresponding author.

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
