# Peer review of "Development and Validation of a Questionnaire of the Perioperative Nursing Competencies in Patient Safety"

_ijerph, 2022, doi:10.3390/ijerph19052584_

Round 1

Reviewer 1 Report

This paper is exciting, and I stand with minor corrections.

  1. How to do a model checking plot? I suggest the author use scaled deviance, AIC, AICc, BIC.
  2. There are many fit index criteria such as SRMR, RMSEA, CFI, TLI, NNFI, X2. I suggest the author do an evaluation using this technique.
  3. Authors need to explain previous studies on SEM at least 30 recent references. Discuss advantages and disadvantages.
  4. The application of SEM is widely previous research is performing various optimization and criteria to find the best parameter on SEMs models like a neural network [1–3], Bayesian [4–6], hierarchical likelihood [7,8], frequentist.

Additional references:

  1. Chong, A.Y.L. A two-staged SEM-neural network approach for understanding and predicting the determinants of m-commerce adoption. Expert Syst. Appl. 2013, 40, 1240–1247, doi:10.1016/j.eswa.2012.08.067.
  2. Binsawad, M.H. Corporate Social Responsibility in Higher Education: A PLS-SEM Neural Network Approach. IEEE Access 2020, 8, 29125–29131, doi:10.1109/ACCESS.2020.2972225.
  3. Ahani, A.; Rahim, N.Z.A.; Nilashi, M. Forecasting social CRM adoption in SMEs: A combined SEM-neural network method. Comput. Human Behav. 2017, 75, 560–578, doi:10.1016/j.chb.2017.05.032.
  4. Assaf, A.G.; Tsionas, M.; Oh, H. The time has come: Toward Bayesian SEM estimation in tourism research. Tour. Manag. 2018, 64, 98–109, doi:10.1016/j.tourman.2017.07.018.
  5. Smid, S.C.; Winter, S.D. Dangers of the Defaults: A Tutorial on the Impact of Default Priors When Using Bayesian SEM With Small Samples. Front. Psychol. 2020, 11, 287–290, doi:10.3389/fpsyg.2020.611963.
  6. Shi, D.; Song, H.; Liao, X.; Terry, R.; Snyder, L.A. Bayesian SEM for Specification Search Problems in Testing Factorial Invariance. Multivariate Behav. Res. 2017, 52, 430–444, doi:10.1080/00273171.2017.1306432.
  7. Caraka, R.E.; Noh, M.; Lee, Y.; Toharudin, T.; Tyasti, A.E.; Royanow, A.F.; Dewata, D.P.; Gio, P.U.; Basyuni, M. The Impact of Social Media Influencers Raffi Ahmad and Nagita Slavina on Tourism Visit Intentions across Millennials and Zoomers Using a Hierarchical Likelihood Structural Equation Model. Sustainability 2022, 14, 1–28.
  8. Caraka, R.E.; Noh, M.; Chen, R.C.; Lee, Y.; Gio, P.U.; Pardamean, B. Connecting Climate and Communicable Disease to Penta Helix Using Hierarchical Likelihood Structural Equation Modelling. Symmetry (Basel). 2021, 13, 1–21.
  9. Jin, S.; Ankargren, S. Frequentist Model Averaging in Structural Equation Modelling. Psychometrika2019, 84, 84–104, doi:10.1007/s11336-018-9624-y.
  10. Jin, S. Essays on Estimation Methods for Factor Models and Structural Equation Models, Uppsala: Acta Universitatis Upsaliensis, 2015.

Author Response

1) How to do a model checking plot? I suggest the author use scaled deviance, AIC, AICc, BIC.

AIC and BIC are calculated from log-likelihoods, so they are available for MLE method only, not least square estimators. We have used model fit index criteria instead.

2) There are many fit index criteria such as SRMR, RMSEA, CFI, TLI, NNFI, X2. I suggest the author do an evaluation using this technique.

The values of the RMSEA, CFI y TLI were previously stated in the article.

We add the value of the SRMR (Standardized Root Mean Square Residual): 0,081. Page 5 line 203-204 & Page 11 line 353 and 357-358.

NNFI (normed fit index): An NFI of 0.95, indicates the model of interest improves the fit by 95\ null model. The NNFI (also called the Tucker Lewis index; TLI) is preferable for smaller samples. For this reason, due to the size of our sample, we have calculated the TLI.

3) Authors need to explain previous studies on SEM at least 30 recent references. Discuss advantages and disadvantages.

Hypothesizing the structural model is essential in SEM, and must be considered sound, at least enough for us to be able to explain a good or bad fit, and/or modification that are performed on the measurement model. When there is no hypothetical structure or the structure of the measurement model (instrument or test) is questionable, an exploratory factorial analysis is preferred, which can be used to find the relationships between the items and/or indicators with the latent variable, through exploration. In our case, we started with a hypothesized model that was constructed through a predetermined structure, according to the accepted conceptual framework, which was the EORNA, and a construct through the use of a three-round modified Delphi method. In this manner, we sought to explore if the model had a good or bad fit between its variables, that is, to assess the variables’ dependence and independence with each other. Also, to assess if these two models were able to make estimations of latent variables through observed variables.

An exploratory factorial analysis was performed, but it classified the structure of factors and variables differently and mainly without sense. The questionnaire follows the order of the perioperative process, and also includes the main activities associated with the safety of the patient according to nursing roles. The classification of the variables into another category does not make sense. Thus, the exploratory factorial analysis was discarded.

Here you find some of the main literatura that  supports our approach and análisis:

  • Yves Rosseel (2012). lavaan: An R Package for Structural Equation Modeling. Journal of Statistical Software, 48(2), 1-36. URLhttps://www.jstatsoft.org/v48/i02/.
  • Awang, Z. (2012). A handbook on SEM. Structuralequationmodeling.
  • Schumacker, R. E., \& Lomax, R. G. (2004). A beginner's guide to structural equation modeling, Second edition. Mahwah, NJ: Lawrence ErlbaumAssociates.
  • Carvajal A, Centeno C, Watson R, Martínez M, Rubiales AS. ¿Cómo validar un instrumento de medida de la salud? [How is an instrument for measuring health to be validated?]. AnSistSanitNavar. 2011 Jan-Apr;34(1):63-72. Spanish. doi: 10.4321/s1137-66272011000100007. PMID: 21532647.
  • Batista-Foguet JM, Coenders G, Alonso J. Análisis factorial confirmatorio. Su utilidad en la validación de cuestionarios relacionados con la salud [Confirmatory factor analysis. Its role on the validation of health related questionnaires]. Med Clin (Barc). 2004;122 Suppl 1:21-7. Spanish. doi: 10.1157/13057542. PMID: 14980156.
  • Brown, T. A. (2015). Confirmatory factor analysis for applied research(Second edition). New York; London: The Guilford Press.
  • Hye-Young J. & Eun-Ok S. (2020). Development and Validation of the Scale for Partnership in Care—for Family (SPIC-F). Int. J. Environ. Res. Public Health 2020, 17, 1882; doi:10.3390/ijerph17061882

We have provided a short explanation, and this has been added to page 5, lines 207-215

4) The application of SEM is widely previous research is performing various optimization and criteria to find the best parameter on SEMs models like a neural network [1–3], Bayesian [4–6], hierarchical likelihood [7,8], frequentist.

The confirmatory factor analysis is a technique that allows assessing the construct validity of measurement scales.  We use the lavaan package of R, which easily allows testing structural equation models. The lavaan is a quality package developed to provide researchers for latent variable modeling. The lavaan can be used to estimate a large variety of multivariate statistical models, including path analysis, confirmatory factor analysis, structural equation modeling and growth curve models. For this reason it was used in our research.

Thank you very much for providing us with all these references in relation to statistical models.

Reviewer 2 Report

Thank you for the opportunity to review, the topic of patient safety is very important 

I suggest you move  Lines 214-225 Strenghs, Limitation, and Areas for Further Research before  Conclusions

I propose to resign Lines 426-428 Patents

Validation correct, developed long questionnaire - weak punt, difficult to fix by rewriting the manuscript 

Author Response

Thank you for the opportunity to review, the topic of patient safety is very important 

I suggest you move Lines 214-225 Strenghs, Limitation, and Areas for Further Research before Conclusions – Done, page 13 lines 447-458

I propose to resign Lines 426-428 Patents – Done, page 13 lines 466 - 468

Validation correct, developed long questionnaire - weak punt, difficult to fix by rewriting the manuscript  - Thank you for your comment. We are aware of the lenght of the questionnare. We are working on a shorter versión to be uses in nurse professionals

Reviewer 3 Report

It was a pleasure to review this manuscript. I believe that the methods are robust, results are clearly presented, and the CUCEQS tool can have meaningful implications for patient safety and outcomes. I have a few comments regarding the introduction and discussion aimed to enhance the manuscript. Please see these detailed comments below.

Abstract: I suggest adding a sentence about the significance of the research

Intro:
The introduction is clear and presents a clear review of the literature and what this study adds to the science.

To enhance the description of the "study motivation," I suggest clarifying why the nursing workforce is the focus of the questionnaire. Starting on line 58, the authors briefly discuss that the questionnaire focuses on nursing, and the authors mention that factors surrounding the multidisciplinary team are linked to post-operative complications. Though the specific link between nurses and surgical patient safety are not clearly and explicitly stated. Please elaborate on this.

Line 40: please clarify: 3-16% of the 25% or of all patients?

Line 49: While most of the audience may be familiar with the "surgical verification list", please briefly explain what it is for those who may be unfamiliar.

Methods:
General: Methods are very clearly described and robust.

Line 105: Were these 5 years of continuous work experience full time, part time, or mixed?

Results:
Well-presented and articulated

Discussion:
Well-written and focused on important aspects of the findings.

I suggest adding a bit more on the significance of the CUCEQS for patient safety and how/if the CUCEQS can be operationalized in work settings.

I'm curious about international implications of this work. Do the authors think that in the future, the tool can be suited to other countries even outside of Europe (i.e., United States, Canada, South Korea)? If so, I suggest including a sentence or two about this in the discussion as it bolsters the implications of this work.

Conclusion:
Clear and fits with findings

Appendices:
Thank you for including appendices. These were helpful to reference.

Author Response

It was a pleasure to review this manuscript. I believe that the methods are robust, results are clearly presented, and the CUCEQS tool can have meaningful implications for patient safety and outcomes. I have a few comments regarding the introduction and discussion aimed to enhance the manuscript. Please see these detailed comments below. Thank you very much for your suggestions

Abstract: I suggest adding a sentence about the significance of the research. Done, page 2 lines 31-35.

Intro:
The introduction is clear and presents a clear review of the literature and what this study adds to the science. Thank you

To enhance the description of the "study motivation," I suggest clarifying why the nursing workforce is the focus of the questionnaire. Starting on line 58, the authors briefly discuss that the questionnaire focuses on nursing, and the authors mention that factors surrounding the multidisciplinary team are linked to post-operative complications. Though the specific link between nurses and surgical patient safety are not clearly and explicitly stated. Please elaborate on this.

The perioperative nurse is found in a multidisciplinary team and is the only health professional who accompanies the patient throughout the entire perioperative process (preoperative, intraoperative and postoperative patient care). The care provided by the nursing professionals must be based on the safety of the patient, and must be the priority. Considering the conceptual framework of the EORNA, the same European association indicates that the competency structure it has worked on needs a greater development of indicators and descriptors that define these competencies. The intention of the present work was to provide a tool that describes the competences of perioperative nursing as a whole, that is, considering their roles, and especially underlining the behaviors associated with the safety of the patient. Until now, a questionnaire with these characteristics was not available.

We have provided a short explanation of the reasons why we focused on nursing personnel, and this has been added to page 3, lines 67-68.

Line 40: please clarify: 3-16% of the 25% or of all patients? Done, page 2 line 44

Line 49: While most of the audience may be familiar with the "surgical verification list", please briefly explain what it is for those who may be unfamiliar.

The checklist is a verification tool created by the WHO, which focuses on five specific areas of the perioperative process correct marking of the surgical area, establish systems to avoid medication errors related to the anesthetic act, correct and systematic counting of gauzes, develop effective communication and minimize interruptions in a cohesive surgical team), with the objective of fulfilling all the items found in each of the areas to reduce the number of adverse errors.

We have provided a short explanation of the checklist tool and this has been added to page 2 lines 54-58.

Methods:
General: Methods are very clearly described and robust. Thank you
Line 105: Were these 5 years of continuous work experience full time, part time, or mixed? The work experience is mostly full time. We have added it to the text. Thank you. Add page 4 line 116

Results:
Well-presented and articulated. Thank you

Discussion:
Well-written and focused on important aspects of the findings. Thank you

I suggest adding a bit more on the significance of the CUCEQS for patient safety and how/if the CUCEQS can be operationalized in work settings.

We consider that your suggestion is expressed in this paragraph (page 12, line 423-426). However, to reflect your suggestion we have modified the original text.  

I'm curious about international implications of this work. Do the authors think that in the future, the tool can be suited to other countries even outside of Europe (i.e., United States, Canada, South Korea)? If so, I suggest including a sentence or two about this in the discussion as it bolsters the implications of this work.

We believe that the perioperative care plans are universal and comparable at the international level, therefore, the indicators of the questionnaire would be international too. But, the difference may exist at the time of defining the roles of perioperative nursing, which are different if we compare them internationally. Even cultural and interpersonal relationship differences could influence. For this reason, we have not added your suggestion to the article.

Conclusion:
Clear and fits with findings. Thank you

Appendices:
Thank you for including appendices. These were helpful to reference. Thank you